# Advances in Vaporized Hydrogen Peroxide Reusable Medical Device Sterilization Cycle Development: Technology Review and Patent Trends

**DOI:** 10.3390/microorganisms11102566

**Published:** 2023-10-15

**Authors:** M. R. Karimi Estahbanati

**Affiliations:** Chemical Engineering Department, Université Laval, 1065 Avenue de la Médecine, Quebec, QC G1V 0A6, Canada; mahmoodreza.karimiestahbanati.1@ulaval.ca

**Keywords:** sterilizer, 510(k), sterilant, medical device, VHP

## Abstract

Vaporized hydrogen peroxide (VHP) terminal sterilization is one of the most promising techniques for sterilizing temperature-sensitive medical instruments like endoscopes. This technique requires only electricity and sterilant containers to perform the sterilization process in less than 1 h without any substantial safety concerns for patients, personnel, and the environment. This review studies recent advances and future trends in VHP sterilization cycle development using U.S. patent applications and 510(k) premarket notifications. In this regard, the patents focused on increasing VHP concentration or handling residual moisture are discussed in depth. The expired patents are analyzed to introduce existing unencumbered technologies, and active patents are presented to show the most current novelties and technology trends. In addition, 510(k) premarket notifications are explored to evaluate implemented technologies in US market-leading commercial products.

## 1. Introduction

The COVID-19 pandemic reminded us of the importance of the continuous improvement of medical technologies to fight against evolving pathogens. Incomplete sterilization of reusable medical instruments can be fatal, and sterilization must be followed properly before instrument reuse on a patient to reduce the probability of infection. Steam sterilization is one of the most common sterilization methods that operates around 121 °C, therefore, cannot be used for temperature-sensitive instruments like endoscopes made of polymeric parts which often cannot tolerate temperatures above 60 °C [1] or surgical instruments with delicate components. Ethylene oxide gas (EOG) sterilization is used mainly for industrial low-temperature sterilization of medical instruments and for hospital applications facing safety concerns for personnel, patients, and the environment, as well as long overall reprocessing time due to extended ventilation. Other chemical agents, like glutaraldehyde and formaldehyde, have been also used, respectively, mainly for high-level disinfection and general disinfection, but their application remained limited because of safety and efficacy considerations [2]. It is worth mentioning that the primary objective of sterilization is to completely eliminate or kill all forms of microorganisms and to make the item entirely free from viable microorganisms, rendering it sterile. Disinfection though aims to only reduce the number of microorganisms to a level that is considered safe for human health. Accordingly, reliable sterilization technologies are required to efficiently and safely sterilize temperature-sensitive medical instruments like endoscopes.

Advanced oxidation processes (AOPs) take advantage of highly reactive species, like hydroxyl radicals, which can rapidly and effectively degrade a wide range of organic and inorganic pollutants, including microorganisms. Though AOPs were first introduced for environmental remediation, they obtained attention in other industries, like the sterilization of medical devices, due to their high efficacy. Liquid hydrogen peroxide is one of the most common AOP agents, but its medical application has been generally limited to antiseptic and disinfectant. Vaporized hydrogen peroxide (VHP) demonstrated higher antimicrobial efficacy than its liquid form and could be used with a concentration as low as 1–10 mg/L [2]. This might be because of different mechanisms of action in the vapor phase [3] as well as its rapid penetration into diffusion-restricted environments in vacuum conditions. VHP sterilization application is growing rapidly due to a new revision of AAMI ST91, which suggests enhancing flexible endoscope reprocessing from high-level disinfection to sterilization [4]. VHP was also combined with other sterilization technologies, like plasma [5] or ozone [6], although, through time, VHP converted into the main sterilizing agent and the initial technology primarily plays the role of safety in removing the remaining hydrogen peroxide at the end of the sterilization process. VHP is compatible with most polymeric materials found in medical instruments like endoscopes. In addition, it demonstrated higher patient and healthcare worker safety in comparison to other gaseous oxidizing agents like EOG since it can degrade easily into water and oxygen. Stringent regulatory compliance, precise monitoring, and comprehensive training further bolster safety, making VHP sterilization a trusted and efficient method for safeguarding patients, personnel, and the environment in healthcare settings. VHP sterilization devices require only electricity and a sterilant container; they sterilize in less than 1 h and require no extended aeration. However, only bone-dry and non-cellulose-based medical instruments are compatible, and there are still compatibility issues for some materials like nylon.

In general, VHP sterilization is effective, safe, compatible, fast, and energy-efficient, particularly when compared to other sterilization methods. VHP’s efficacy is attributed to its ability to rapidly penetrate even diffusion-restricted environments, effectively killing microorganisms. Compared to traditional methods like steam sterilization, which cannot be used for temperature-sensitive instruments like endoscopes, VHP offers a versatile solution. It excels in achieving a high level of microbial reduction while also being compatible with most polymeric materials. Furthermore, VHP’s non-toxic by-products and minimal environmental impact make it a safer choice. Compared to alternatives like EOG, which can leave hazardous residues and require extended ventilation periods, VHP’s quick sterilization cycle and generation of nontoxic residues (water and oxygen) enhances both instrument safety and workflow efficiency. Furthermore, VHP can complete the sterilization process in less than one hour, making it notably faster than methods like steam sterilization, which require longer cycles, or EOG sterilization, which requires ventilation for several hours. Additionally, VHP operates at lower temperatures, reducing energy consumption and making it an energy-efficient choice in comparison to steam sterilization, which requires evaporation of extensive amounts of water, and EOG, which needs several hours of ventilation. On the other hand, due to significant constraints like scale, penetration, and compatibility with packaging materials, the widespread adoption of VHP sterilization for single-use devices has yet to obtain traction. Nevertheless, recent advancements in the sterilization chamber design and cycle development offer a fresh perspective and potential for exploration [7].

VHP sterilization for medical devices is subject to rigorous regulatory requirements, particularly in the United States, overseen by the Food and Drug Administration (FDA). Compliance involves adhering to recognized standards such as ANSI/AAMI ST58:2013 [8], and entails a meticulous validation process. Manufacturers must conduct validation studies demonstrating consistent microbial inactivation using biological indicators and routine monitoring of critical parameters. Continuous monitoring and validation are necessary to accommodate changes in cycle design, device design, and materials, ultimately safeguarding the safety and effectiveness of medical devices treated with VHP sterilization in healthcare settings.

Several VHP sterilization family devices have been approved by the FDA for sale in the US market, including STERRAD (Advanced Sterilization Products (ASP), Irvine, CA, USA), V-Pro (Steris, Inc., Mentor, OH, USA), PSD-85 (Sterilucent Inc., Minneapolis, MN, USA), and STERIZONE VP4 (Stryker Inc., Kalamazoo, MI, USA). Currently, STERRAD and V-Pro are the most widely used sterilizers in the US market. STERRAD 100NX and V-PRO maX are larger capacity hospital sterilizers, while STERRAD NX and V-PRO S are smaller devices with a lower chamber volume. From a hardware point of view, the main differences between STERRAD and V-PRO sterilization cycles are related to plasma generation, VHP concentration increase, and VHP concentration measurement. The cycles available in a single sterilizer model could only use a part of the hardware. For example, only STANDARD and FLEX cycles of STERRAD 100NX use a VHP concentrator, and only the DUO cycle uses a delivery module to temporarily hold hydrogen peroxide from a cassette and dispense a smaller amount of sterilant during each half-cycle.

In VHP sterilization devices, the sterilization cycle is typically composed of two identical half-cycles. The use of these half-cycles is often based on a method known as the “Overkill Approach”, as described in the ISO14937:2009 standard [9]. In this approach, a routine sterilant dwell period is extrapolated (to 12 spore log reduction) from the time required to achieve a 6-log reduction in a population of 10^6^ biological indicators (BIs), ultimately providing a Sterility Assurance Level (SAL) of ≤10^−6^. This method assumes that the microbial inactivation kinetics follow a linear pattern, similar to what is observed in EOG sterilization of BIs like *Bacillus atrophaeus* (*B. atrophaeus*, ATCC 9372), as noted in ISO 11138-2 [10] for EOG. It relies on the assumption that microbial populations treated with VHP will exhibit the same resistance to the applied lethal stress, as represented by a log-linear inactivation kinetic plot, often using populations like *Geobacillus stearothermophilus* (ATCC 7953) or *B. atrophaeus* as Bis [11].

The VHP sterilization half-cycle usually consists of three main steps, namely conditioning, sterilization, and venting. The conditioning step aims to remove air from the load to facilitate the penetration of VHP as well as remove traces of moisture remaining on the load. The pressure at the end of the conditioning step is less than 1 Torr and in STERRAD sterilizers could produce plasma for up to 15 min to enhance moisture removal. During the sterilization step, which is the main part of the process, VHP is injected in a fixed amount (STERRAD and V-pro) or fixed differential pressure (STERIZONE VP4). Since the obtained pressure is higher than the dew point of hydrogen peroxide solution at ambient temperature (around 5 Torr [12,13]), a microlayer condensate forms on the load. This means that, contrary to popular belief, VHP condenses inside the chamber in these sterilizers. To enhance the sterility inside a diffusion-restricted environment, like lumens, the air is usually injected after VHP to deliver it into diffusion-restricted environments. The sterilization phase is repeated one or a few times to obtain an additional 6-log reduction of the resistant bacterial spore. During the venting step, which aims to enhance safety, the chamber is evacuated to transfer VHP to a catalytic converter and is then filled with fresh filtered air. As an example, cycle parameters of the Flexible cycle of V-PRO^®^ maX 2 sterilizer are extracted from its Technical Data Monograph [14] and summarized in Appendix A. In general, the configurations of STERRAD and V-Pro sterilization cycles are similar, with differences in the conditioning pressure (0.15–1 Torr), plasma time (0–15 min), amount and concentration of injected VHP (59–96%), the number of VHP injections (2 or 4), VHP post-injection pressure (6–14 Torr), duration of hold after VHP injection (1.5–6 min), after air injection (0.75–10 min), and target air injection pressure (760 or 500 Torr) [14,15,16,17]. To the best knowledge of the author, no review has been performed on the VHP sterilization cycle development.

In this work, the current developments and future trends of cycle development for VHP terminal sterilization are reviewed. In this way, special focus has been given to patent applications and 510(k) premarket notifications as the most reliable sources of information published by manufacturers.

## 2. Analysis of Patent Trends

In this section, U.S. patent applications related to low-temperature VHP terminal sterilization are reviewed. Expired patents are discussed to demonstrate existing unencumbered (patent-free) technologies, as well as active patents to show the most current novelties and technology trends. Unencumbered technologies are free from any existing intellectual property rights, such as patents, allowing for unrestricted use and development. This section could be a guide for research trends on VHP sterilization cycle development and potential game-changing prospective technologies and features in the next generation of sterilizers.

The examination of U.S. patent applications on VHP sterilization revealed that most of the patents in this area aim at increasing VHP concentration, handling residual moisture, compatibility improvement, integration with other technologies, energy efficiency improvement, personnel safety enhancement, and environmental safety enhancement (Figure 1). Particularly notable inventions include precise control of VHP concentration and inventive moisture removal techniques, and underscoring a collective commitment to optimizing sterilization processes’ efficiency and safety. In addition, the patents on this category generally claim modifications in the sterilization *cycle*, which is the main aim of this review. Therefore, this review is focused on the patents aim to enhance VHP concentration and residual moisture handling.

To enhance the antimicrobial efficacy, some patents claim technologies to increase VHP concentration in the chamber, while others specifically target its increase inside diffusion-restricted environments like lumens. On the other hand, the patents focused on handling residual moisture address the detection of residual moisture and/or its elimination.

### 2.1. Increasing VHP Concentration

The antimicrobial efficacy of the sterilization process depends on the sterilant contact with microorganisms on the load. To increase the contact, a higher amount of sterilant could be injected, however, this approach causes some issues in the practice. For example, this could impact not only the compatibility of some instruments, but also their safety because of a higher amount of remaining absorbed or adsorbed sterilant on the load after sterilization. Moreover, this is preferred to remain close to the vapor phase condition as the literature suggests VHP sterilization is more effective than its liquid phase [3]. In addition, by injecting a higher amount of VHP, its concentration inside diffusion-restricted environments, like long lumens, could not necessarily increase, because hydrogen peroxide preferentially condenses outside the lumen, and water-rich vapor is transferred into the depth of the lumen. A thorough cost–benefit analysis and consideration of specific application requirements are essential in determining the economic feasibility of increasing VHP concentration in a sterilization cycle. Another approach to enhance the VHP contact with microorganisms is increasing the concentration of hydrogen peroxide in the vapor phase (for example from 59% to around 90% [14,15]). This could be particularly effective to increase the VHP concentration inside diffusion-restricted environments like long and narrow lumens, which may be as long as 3.5 m and have an internal diameter of 0.7 mm. Sterility of diffusion-restricted environments is critical as a worst-case scenario that affects the compatibility and safety of sterilized medical devices. Figure 2 summarizes different approaches to increasing the concentration of VHP inside the chamber or inside the lumen. The approaches to increase the concentration in the chamber could potentially increase the concentration inside lumens.

#### 2.1.1. Inside Chamber

Increasing the concentration of VHP inside the sterilization chamber is important since it could significantly affect the concentration inside the lumens. Increasing the concentration of hydrogen peroxide in its liquid aqueous solution is a method to increase the concentration of VHP in the chamber. Hydrogen peroxide is thermodynamically unstable, especially in the presence of most impurities. However, its concentration could be increased using stabilizers like sodium stannate in the presence of soluble pyrophosphate or a phosphate-pyrophosphate mixture [18]. Because of technical issues during storage, transportation, and application of hydrogen peroxide sterilant solutions, sterilant concentration is regularly no higher than around 59% [14,15].

VHP condensation inside the chamber affects its concentration since hydrogen peroxide condensates are preferentially compared to water. The condensation could not be avoided in many cases and affects the concentration distribution inside the chamber, i.e., water-rich VHP is transferred into diffusion-restricted areas, which makes their sterilization more difficult. Laflamme et al. [13] developed a method to detect VHP condensation using pressure data analysis. They suggested pressure monitoring during the VHP injection step as a method to detect condensation occurrence or degree of condensation. This monitoring could detect a change in the rate of pressure increase or deviation/degree of deviation/amount of deviation from a theoretical pressure. This method is non-invasive, does not require direct measurement of load temperature, and takes into account load temperature variation during conditioning or previous sterilant injection phases. The inventors showed the detected dew point of this method at 18, 25, and 30 °C were close to their theoretical values of around 3, 5, and 7 Torr. Accordingly, this approach could help in the detection of the dew point to avoid further sterilant injection and therefore decrease the concentration of VHP inside the chamber.

One approach to increase the concentration of hydrogen peroxide inside the chamber is concentrating hydrogen peroxide before its injection into the chamber. ASP developed NX™ technology to concentrate 59% hydrogen peroxide using an evaporation and condensation system. For example, the Standard and Flex cycles of STERRAD 100NX concentrate the sterilant up to approximately 90% nominal hydrogen peroxide just before its injection into the chamber by its selective evaporation and then condensation in a condenser [15]. Jacobs et al. [19] developed a technology to concentrate hydrogen peroxide, and Kolher et al. [20] improved this technology by increasing the speed of the sterilization process, especially for loads having a lumen. Ahiska [21] introduced an alternative injector-concentrator arrangement, which is expected to provide better control of the concentration of hydrogen peroxide. The concentration of VHP inside the chamber could be measured using different technologies such as spectrophotometrical [22] or electrochemical [23] methods.

The concentration of hydrogen peroxide before its injection into the chamber requires specialized technologies. An approach to increase the concentration of hydrogen peroxide could be its condensation on the load and then reducing the chamber pressure to evaporate the formed condensate [24]. In this method, VHP must be injected up to a pressure higher than the dew point of the sterilant (depending on the load temperature) to form condensate on the load. After pressure reduction, since water preferentially evaporates at a reduced pressure, water-rich vapor could be transferred out of the chamber to be replaced with a hydrogen peroxide-rich vapor, which evaporates from the remained condensate. Although this method could be beneficial in some cases to improve antimicrobial efficacy, compatibility and safety issues could occur. Wang et al. [24] showed that inside long lumens could not be sterilized by increasing the amount and duration of VHP exposure; however, sterilization could be achieved by injection of VHP up to around 9.5 Torr to condense the sterilant followed by an evacuation to evaporate it. They also showed that sterility could be enhanced by venting the chamber to an atmospheric pressure to transfer the concentrated VHP into the lumen as well as by performing multiple condensations and evaporations. In another work, condensed hydrogen peroxide on the load was evaporated using an unsaturated gas to increase its concentration [25].

The concentration of hydrogen peroxide in the chamber could be reduced after its injection because of its adsorption on the chamber walls and continuous degradation of a part of that. This could have a significant effect, especially for a small chamber that has a high surface area over volume ratio as well as for a chamber made of relatively high hydrogen peroxide adsorptive materials like aluminum. The adsorption could be significantly higher in the first half-cycle, therefore, the sterilant could be injected in the second half-cycle more than the required amount to achieve a 10^−6^ SAL. One approach to address this issue could be prime injection, i.e., to inject sterilant into the chamber and then evacuate to a vacuum pressure during the conditioning step (Figure 3) [26]. The saturation of adsorption sites after the prime injection could be close to its condition just before the second half-cycle. The inventor showed that the maximum concentration of hydrogen peroxide inside the chamber was lower in the first half-cycle in a sterilization cycle without a prime injection; however, the concentration was equal in the first and second half-cycles for a process taking advantage of a prime injection. The amount of injected sterilant during prime injection could be around 10–30% of the amount injected in a typical sterilization cycle. This technology is employed in all the cycles of V-PRO^®^ S2 [16]. Though useful for sterilizers with a small chamber volume, like V-PRO^®^ S2, this approach could not make a significant difference for sterilizers with a large chamber volume like V-PRO^®^ maX 2. This could be a reason for removing prime injection from the Lumen cycle in the next generation of V-PRO^®^ maX (V-PRO^®^ maX 2) [14].

#### 2.1.2. Inside Diffusion-Restricted Environments

The concentration of hydrogen peroxide inside medical device lumens could be significantly less than its concentration inside the chamber because of different phenomena. Water has a higher vapor pressure than hydrogen peroxide, which means it preferentially evaporates from liquid sterilant. Based on this thermodynamic property, hydrogen peroxide could preferentially condense outside lumens after sterilant injection into the chamber and the concentration of VHP reduces continuously during penetration through lumens. Furthermore, water has a higher diffusivity than hydrogen peroxide since it is a smaller and lighter molecule. As a result of this mass transfer property, water diffuses faster than hydrogen peroxide inside lumens; therefore, the concentration of VHP reduces even more during penetration through lumens. In addition, hydrogen peroxide is prone to decompose into water and oxygen through its path to lumens. Moreover, the remaining air at the vacuum (<1 Torr) inside the lumens is trapped and compressed at the center of the lumen. At this location, the trapped air decreases the antimicrobial efficacy because of significant VHP dilution. Based on these, the concentration of hydrogen peroxide inside the lumen could be significantly less than in the chamber. Therefore, this is essential to develop strategies for increasing the concentration of hydrogen peroxide inside lumens as a worst-case sterilization condition.

Some approaches for lumen VHP concentration enhancement are well-known in the industry. As mentioned above, water has a higher vapor pressure than hydrogen peroxide, which means it evaporates preferentially from the evaporator and enters the chamber first. As a result, later vapors, which are rich in hydrogen peroxide, push and transfer initial water-rich vapor into lumens. To avoid this problem, hydrogen peroxide could be transferred into the evaporator in micropulses to feed an almost uniform VHP through the sterilant injection step. In addition, the chamber pressure is reduced to less than 1 Torr before VHP injection to facilitate delivering VHP into lumens. This approach avoids the compression of the remained air in the middle of lumens. Childers et al. [27] showed the depth of VHP penetration into a 1 × 120 cm (ID × Length) lumen could increase from 67% to 96% by reducing the pressure from 5 to 0.1 Torr.

A proven approach to increase the concentration of VHP inside lumens is delivering VHP from the chamber into the lumen using a non-condensable gas like filtered air. In this approach, VHP is first injected into the chamber following a hold time to develop a uniform distribution. A non-condensable gas is then injected into the chamber up to as high as atmospheric pressure to transfer VHP into lumens following a hold time to permit sufficient contact time between VHP and microorganisms. The injected gas could also significantly reduce the area in the middle of the lumen that is in contact with the trapped compressed air. The injected gas could also form fluid turbulences and increase the concentration of VHP at the compressed region. On the other hand, the injection of gas causes complexities in the sterilization process. For example, this could change the phase equilibrium because of modifying parameters like gas temperature and relative humidity. Childers et al. [27] showed the depth of VHP penetration into a 1 × 120 cm (ID × Length) lumen at 1 Torr could enhance from 73% up to 98% by vapor compression up to 165 Torr. Nowruzi et al. [28] suggested a step-wise injection of non-condensable gas with a hold time between them as a method to enhance the penetration of VHP into lumens (Figure 4). This method better agitates chamber content by reducing the formation of airflow channels that go directly from the air injection inlet into the packaging/lumen. An efficient step-wise injection could consist of incremental pressure increase to targets, which increase exponentially, to be effective at low pressures and fast at high pressures.

One approach to increase the concentration of VHP inside lumens is using a vessel attached to one side of the lumens. The vessel could be either empty or filled with liquid or solid (urea peroxide complex or sodium pyrophosphate peroxide complex [29]) sterilant. The empty vessel, having at least two times of lumen volume, plays the role of a space that sucks VHP into the lumen during the VHP injection step and also hinders compression of remaining air at the center of the lumen [30]. The filled vessel generates VHP and pushes out all the air inside the lumen at the end of the evacuation step. This could ensure that all the lumen is in contact with high-concentration VHP, decrease the duration of the sterilization cycle, and enhance material compatibility. However, in addition to the safety concerns, sterilization of the contact area between the lumen and the vessel is very challenging. Wu et al. [31] developed a technology to have minimal contact between the vessel and lumen (as low as the contact area of a scissor), however, this also could not guarantee the sterilization of the contact area. This has been a major challenge in employing the STERRAD booster and adaptor, which was used for the STERRAD 100S System introduced in 1993. In addition, this technique could significantly increase the probability of residual liquid left over at the end of the sterilization cycle.

Recently, some other inventors proposed concepts to inject VHP generated from an evaporator directly into lumens, rather than the chamber. For example, Conseil et al. [32] developed a concept to directly inject VHP into lumens using tubes connecting them to the sterilizer’s VHP injection port (Figure 5). Their concept takes advantage of some specialized connectors installed on the chamber walls as well as packaging to develop a fluid connection between the injection port and lumen. Deprey et al. [1] proposed placing extra evaporators within the chamber and connecting them directly to one side of the lumen. In this way, VHP is generated by primary sterilizer evaporators and sterilizes outside lumens and the extra evaporator then sterilizes inside lumens. The inventors claimed this technology is effective in sterilizing 1 mm diameter lumens with a length as high as 3.5 m. All these approaches could also suffer from the sterility of the instrument’s contact area with the tubes transferring sterilant into them.

### 2.2. Handling Residual Moisture

Before VHP sterilization, the instruments need to be cleaned in a water and detergent bath and then completely dried following the manufacturers’ instructions. Although medical facility personnel dry the devices using methods including compressed air, heat, and towels, residual moisture could remain before the load packaging. Failure to thoroughly dry the instruments could lead to some issues like difficulty in reaching the target vacuum pressure, load non-sterility, and post-sterilization residual hydrogen peroxide on the surface of the load. The sterilization process could tolerate a low amount of residual moisture since it evaporates during the conditioning vacuum. However, a high amount of remaining moisture on the load continues evaporating during the vacuum, hinders effective pressure reduction, and could even cause cycle cancellation. The evaporated moisture takes the latent heat of evaporation from the remaining liquid and reduces its temperature up to ice formation. The formed ice could lead to load non-sterility because of hindering efficient contact of the sterilant with the microorganisms on the load, as well as blocking narrow lumens [33]. In addition, the concentration of sterilant in the gas phase could be reduced as a result of its condensation on the cold spot as well as evaporation of the formed ice. This concentration reduction could even affect the sterility of dry spots, especially in diffusion-restricted environments. This is the reason that VHP sterilizer manufacturers usually only claim the sterility of dry instruments. It is worth mentioning that the formed ice can pop out and be transferred to other locations. On the other hand, depending on the load material (thermal conductivity) and cycle condition, extensively condensed sterilant could remain at the end of the sterilization process and impose health risks for users and patients. Therefore, this is essential to check for the presence of moisture on the load, either during conditioning or venting stages, and perform required actions like cycle abortion or moisture elimination.

#### 2.2.1. Residual Moisture Detection

Handling residual moisture in VHP sterilization is paramount as it ensures the effectiveness and safety of the sterilization process. This is essential because moisture can impede the penetration of VHP, potentially leading to incomplete sterilization and posing risks to patient safety. Moreover, moisture can act as a protective shield for microorganisms, making effective sterilization challenging. Detecting moisture in a VHP sterilization cycle presents several challenges. Residual moisture, often hidden in diffusion-restricted areas of medical instruments, can interfere with the sterilization process, making accurate detection crucial. However, achieving precision in moisture measurement, especially in real-time, can be technologically demanding. Additionally, instrument design and material compatibility must be carefully considered to ensure reliable moisture detection. Innovative monitoring technologies continue to evolve to address these complexities in moisture detection.

Some sterilization systems check the presence of initial residual moisture before introducing VHP to remove them or abort the sterilization cycle. This could be performed by analyzing the pressure of the chamber during/after the conditioning vacuum. Any residual moisture on the load evaporates rapidly at low pressures and changes the rate of chamber pressure reduction or even increases the pressure. If the amount of moisture is non-tolerable, the cycle could abort and ask for complete drying of the load by the user to avoid performing a complete cycle that could lead to non-sterility. Otherwise, the sterilizer either operates normally or triggers an additional step to remove the detected moisture.

Developing a reliable technology to accurately detect initial residual moisture is challenging as this could have different influences depending on the load material, load configuration, and sterilization cycle. During the preconditioning vacuum, the moisture could be detected by the sterilizer but turned into ice at the end of the vacuum. Therefore, after triggering the elimination step, the formed ice could remain in the solid phase and not be detected by some technologies that rely on the chamber pressure analysis. This can be partially addressed by developing technologies to estimate the amount of initial residual moisture that could generate a high amount of ice. Since the formation of ice depends on the load material, developing technologies to estimate the amount of simultaneously present moisture on different material surfaces is complex [33]. For example, metallic instruments have a higher thermal conductivity and retard ice formation by transferring heat from the load into residual water. On the other side, the residual water on a polymeric instrument tends to form ice more easily. The formed ice could remain at the end of the sterilization process and even form residual liquid on the surfaces that were originally dry.

Different patents rely on the analysis of chamber pressure variations to detect the initial residual moisture. This approach is practical since it requires no/minimal hardware modifications and could be performed/enhanced by software upgrading. For example, some systems check for a pressure increase in conditions where it is expected to decrease continuously [12,33] or remain constant [34]. Some of these approaches could face limitations, especially in the presence of a small amount of moisture because of being affected by other parameters. For instance, a virtual leak from an instrument or packaging could simulate the same effect as the initial presence of moisture on the pressure data. A virtual leak is a situation in which the chamber appears to have a leak, but in reality, it is caused by outgassing, which is the release of gas from the surfaces of the chamber or load. Table 1 summarizes different approaches in the prior art to detect the presence of initial residual moisture based on chamber pressure analysis, as well as their advantages and limitations.

In the presence of moisture, the required time for the vacuum pump to reach a target pressure is higher because the moisture continuously evaporates and increases the chamber pressure. Therefore, comparing the pump downtime could be a simple method to detect moisture on a load. However, this method could be beneficial only for a known type of load, which makes it almost non-practical for clinical use because of variable factors like virtual leak and load surface area (adsorbed gases). That is why Sheth and Upchurch [35] developed a method to estimate the effect of other factors on pump downtime and eliminate it. In the first step of their method, the required time to reduce the pressure up to the first predetermined pressure target, which is above the saturation pressure of water, is determined. This time depends on factors like type and material of load, type and material of packaging, vacuum pump capacity and service life, local electricity characteristics, sterilization chamber volume and material, system leakage, and partially the amount of water on the load. In clinical use, the load type is the most important varying parameter and can significantly affect pump downtime, mainly because of virtual leaks. In the second step, the required time to reach a second predetermined pressure target which is below the saturation pressure of water is determined. Since water evaporates very fast around its saturation pressure, this time depends on the above-mentioned parameters as well as the amount of water on the load. The difference between these two measured times could be an indication of the presence of moisture after being compared with a threshold defined for a dry system. This approach could still be sensitive to parasitic parameters, like virtual leak, since their effect could not equally distribute during conditioning vacuum and could even exist after reaching the target pressure. This reduces the effectiveness of this approach to detect a small amount of initial residual moisture. As another claim, the inventors developed a partially similar approach to analyze the temperature of gas instead of evacuation time. The idea is based on the reduction of vapor temperature by evaporation of water. In this approach, the minimum gas temperature is compared with a reference dry system temperature (or a system with an acceptable amount of moisture), and a difference could be an indicator of the presence of moisture. In addition to not being sensitive to a low amount of moisture, both of these approaches may not be capable of distinguishing the presence of moisture on high-conductive (metallic) or low-conductive (polymeric) surfaces, which is critical for predicting the probability of ice formation.

Residual moisture in the chamber could suddenly evaporate around the triple point during the chamber evacuation and cause a temporary increase in the chamber pressure. Witte and Eulogio [12] claimed the chamber pressure increase at pressure levels below 5 Torr (triple point of water) as an indicator of the presence of initial residual moisture. At this point, residual water promptly forms ice crystals by releasing latent heat and causing the sudden evaporation of the neighboring portion of water. This phenomenon can cause a sudden pressure increase in the chamber during the conditioning vacuum, which can appear in the form of a local maximum in the pressure versus time graph. For instance, a pressure increase or cumulative pressure increase up to 50 millitorrs at a pressure near 5 Torr could be an indicator of the presence of initial residual moisture. The inventors showed that 3 mL of water divided among four locations formed four separate turning points on the pressure versus time plot, while the same amount of water makes a single large turning point when it is placed in one location. They believe this method is more sensitive to moisture inside the packaging rather than on the surface of the packaging. The turning points obtained by this approach are expected to be significantly less affected by virtual leaks and could potentially detect smaller amounts of water. In this approach, to obtain a reliable result and make a running average, the pressure could be recorded in at least 0.1 s intervals around the pressure expecting to observe turning points (approximately the triple point of water).

McLaren et al. [34] claimed another approach, during which, the pressure is reduced to a target vacuum pressure and the quantity of water vapor over a defined time (preferably between 20 and 120 s) is monitored by a combination of pressure and concentration sensors. The increase in the pressure could be an indication of the evaporation of liquid. To avoid detecting a virtual leak as an indicator of residual moisture by mistake, water and hydrogen peroxide sensors could be used to ensure the pressure increase is related to water vapor. If the amount of residual liquid is higher than an acceptable first threshold (and lower than the second threshold), the cycle could proceed, and if it is even higher than the second threshold, the cycle could abort. With this approach, the concentration sensor must have a low limit of detection of water vapor because of the high volume of the chamber. In addition, it would not be practical if ice is formed or liquids, other than water (like alcohol), are on the load. The inventors claim this method can be used during a slow-pressure reduction to avoid ice formation; however, this could not be possible if a high amount of water is present on a non-conductive (polymeric) load.

The most recent patent in this area claimed using the second-derivative value of pressure versus time at a pressure less than approximately the triple point pressure of water [33]. This method can not only identify a pressure increase during the evacuation, but also detect a variation in the rate of pressure reduction as an indicator of the presence of initial residual moisture. The detection approach of this patent sums positive differences between consecutive second-derivative values and then compares the summation to a threshold to determine whether residual moisture is present on the load. The turning point detected using this approach could appear at different pressures depending on the load material. For instance, the droplets on a metallic load could form a turning point at 5–30 Torr, while for a polymeric load, this pressure could be less than approximately 5 Torr (triple point of water). This means, depending on the amount of residual water, turning points near 5 Torr could indicate the possibility of ice formation and the necessity of aborting the cycle and manually drying the load. The inventors believe residual water between 1–5 mL could be removed by load conditioning; however, water contents higher than 5 mL could be impossible to eliminate. They obtained the thresholds by conducting some tests with different amounts of water as droplets on some test samples, and claimed these thresholds may be valid for the moisture within the load in the form of puddles, tube blockages, or sheets. They also made control tests without droplets to analyze the noise floor (noise inherent in the pressure transducer) of second-derivative values and filter non-reliable turning points. The inventors suggested the development of a feedback system using this approach to discover the instruments prone to repetitive residual moisture detection or personnel that routinely fail to sufficiently dry instruments. Because of the nature of numerical differentiation, in this approach, in addition to pressure data recording frequency, the resolution of data must be high enough to precisely detect turning points. In addition, this approach could not be effective for a small amount of moisture inside a packed lumen because the high resistance towards the chamber environment could muffle the residual water evaporation effects.

#### 2.2.2. Residual Moisture Elimination

Some actions could be performed to remove the residual moisture during either the conditioning or venting stage. These actions could be performed as a part of the sterilization cycle or be triggered as an extra action if moisture is detected. In addition to residual moisture elimination, these actions could enhance safety by reducing the amount of adsorbed sterilant on the load at the end of the process. High-wall temperature, which originally aims at avoiding/reducing the formation of condensate on the walls, causes an increase in load temperature and evaporation of residual moisture. Since the wall temperature cannot increase higher than around 60 °C to avoid damaging sensitive polymeric instruments in the load, the moisture evaporation by this method could be too slow for a practical application. On the other hand, high-wall temperature causes an increase in the temperature of the load during the first sterilization half-cycle. This means that, in the presence of residual water, the initial temperature of the load and, therefore, the amount of formed micro-condensate on it could be different in the later half-cycle, which is against the manufacturers’ claims of two identical half-cycles in their sterilization cycles. Therefore, the development of reliable technologies to remove detected residual moisture is essential.

Following the detection of residual moisture, the sterilizer could simply abort the cycle, ask for load drying, and start a new cycle. Some sterilizers attempt to estimate the amount of residual liquid and determine if it can be removed. The detection and removal could occur either during the conditioning or venting step, and the procedure can be as simple as refilling the chamber with fresh ambient air and re-evacuation to transfer out the residual moisture [34]. Some sophisticated approaches, like incorporating air plasma [36], ultrasonic wave vibration [37], injecting warm and/or dry air [34], and moving air inside the chamber [38], could be employed.

Air plasma is a state of matter that can be produced through the action of magnetic and electric fields. As an excess or routine endeavor, air plasma can be used to assist in the evaporation of residual moisture on the load by increasing the gas phase temperature and then the moisture temperature [39]. As an example, the DUO cycle of STERRAD 100NX sterilizer routinely uses 15 and 2 min of plasma during the conditioning and venting steps, respectively. Around seven times more plasma duration for conditioning than the venting step indicates the importance of air plasma in removing initial residual moisture compared to removing remaining hydrogen peroxide degradation in their technology. The plasma operates at a pressure as low as 0.2 Torr, which could increase up to 1 Torr at the end of air plasma due to the evaporation of liquids as well as increasing the vapor temperature. This plasma stage imposes some limitations in load compatibility and could decrease the service life of some instruments [36].

The initial moisture on the load at the beginning of the sterilization process could not necessarily be the source of the remaining residual moisture at the end of the process. The residual moisture could form during the sterilization process through extensive sterilant condensation because of the high ratio of injected sterilant over load weight/surface area or low-load temperature as well as the capillarity force (trapping micro-condensate in cavities). Therefore, regardless of the presence of residual moisture at the beginning of the cycle, moisture detection/removal could be performed during the venting step. One approach to enhance the removal of residual moisture could be a few minutes of hold time at a higher pressure to increase heat transfer from wall to load [36]. Another approach could be re-evacuation and holding at vacuum pressure for a few minutes to take advantage of a higher rate of evaporation at low pressure [36]. A combination of these approaches, i.e., a hold at atmospheric pressure followed by a hold at low pressure could enhance moisture removal [36]. These approaches could also be combined with other methods like air plasma at a lower pressure [36] to improve the evaporation rate. Another claimed approach to remove residual moisture at the end of the sterilization cycle is using warm air during venting (Figure 6) [40]. The air heating could be performed using the sterilant vaporizer if its heating capacity is enough to significantly increase the air temperature in a reasonable time.

## 3. Analysis of 510(k) Premarket Notification

A 510(k) premarket notification is a premarket submission made to the U.S. Food and Drug Administration (FDA) to demonstrate that a medical device is safe and effective, that is, substantially equivalent to a legally marketed device. Section 510(k) of the Federal Food, Drug, and Cosmetic Act requires device manufacturers to notify the FDA at least 90 days in advance of their intention to market a medical device in the United States. The 510(k) submission must include detailed information about the medical device, its intended use, and any similarities or differences to other devices already on the market. While the analysis of active patents provides a perspective on future potential advances in the VHP sterilization industry, studying 510(k) of FDA-cleared sterilizers is one of the most reliable sources of information about implemented technologies in commercial sterilizers. The 510(k) process primarily concerns devices intended for the U.S. market and it offers valuable insights into the evolving landscape of commercial sterilizers. The data obtained from 510(k) premarket notifications can shed light on the continuous improvement efforts undertaken by manufacturers to enhance user experiences and maintain the highest levels of quality assurance. This should be noted that 510(k) premarket notification may not be representative of medical devices sold outside the US, and national clearances could be studied to obtain information.

Appendix A summarizes 510(k) premarket notification clearances of STERRAD 100NX and NX, as well as V-Pro maX and S series sterilizers. Analysis of this table could clarify cycle modifications of cleared sterilizers. This table reveals most of the notifications extend the previous claims or aim in improving user experiences like software upgrades, graphical user interface enhancement, and network connectivity. No changes have been observed on the cycles after their clearance, which could be attributed to the enormous efforts required for verification and validation activities for numerous medical instruments. The only significant observed modification on the cycle is related to the addition of a load conditioning feature as a part of ALLClear Technology of the STERRAD^®^ 100NX Sterilizer. This feature, which is only available in its Standard, Flex, and Express cycles, “reduces canceled cycles by performing load and system checks and executing a load conditioning step before starting a sterilization cycle” [38]. Since ALLClear Technology changes only the conditioning step and terminates cycles with problematic loads, it is claimed that the feature does not modify the existing sterilization cycles.

## 4. Conclusions

The development and improvement of sterilization technologies are crucial in reducing patient infections caused by evolving pathogens. VHP sterilization has emerged as a more reliable option than EOG sterilization for non-industrial applications involving temperature-sensitive medical instruments. Extensive patent research on VHP sterilization has focused on enhancing VHP concentration within the sterilization chamber or in diffusion-restricted environments, as well as addressing residual water concerns through detection or elimination methods.

Innovative approaches to increase VHP concentration inside the chamber have included techniques such as pre-concentration of VHP before injection, prime injection, and condensation on the load followed by re-evaporation. Patents targeting increased VHP concentration within diffusion-restricted environments, like lumens, have explored strategies such as air injection, stepwise air injection, attachment of lumens to empty or sterilant-containing vessels, and direct injection of VHP into lumens. Additionally, patents addressing residual moisture detection have utilized pressure monitoring during pressure hold, gas concentration monitoring, evacuation time differences, gas temperature differences during conditioning vacuum, local pressure maxima versus time diagrams during conditioning vacuum, and local maxima in the second derivative of pressure versus time diagrams during conditioning vacuum. Furthermore, patents focusing on residual moisture elimination have incorporated technologies like air plasma, ultrasonic wave vibration, and the injection of hot and/or dry air.

Analysis of 510(k) premarket notifications has revealed that many of these notifications aim to extend previous claims or enhance user experiences through software upgrades, graphical user interface improvements, and network connectivity. While these modifications may not directly impact the efficiency of sterilization cycles, they contribute to improved sterilization practices by reducing the likelihood of user errors and providing tools for more efficient supervision and management of tasks. Notably, ASP has introduced significant improvements to the VHP sterilization cycle in recent years, including NX technology for VHP concentration before injection and All Clear Technology for load and system checks, as well as the execution of a load conditioning step prior to initiating a sterilization cycle. While new VHP cycles have been introduced as supplementary options with slight parameter modifications (such as target pressure or duration of hold) to minimize risks associated with developing entirely new products, this approach can limit inventive thinking and impede fundamental improvements. It is essential to invest in in-depth research to obtain a better understanding of the complex simultaneous phenomena that occur during VHP sterilization, including phase equilibrium, fluid mechanics, heat transfer, mass transfer, and reaction kinetics under various environmental and load conditions. This review can pave the way for novel VHP sterilizers, enabling significant advancements.

Future research efforts should focus on developing technologies that increase the concentration of hydrogen peroxide inside diffusion-restricted environments like narrow and long lumens. This not only has the potential to expand the range of medical devices suitable for sterilization by VHP systems, but also reduce cycle duration, improve material compatibility, and minimize safety concerns. To achieve this, alternative pressure alteration patterns should be explored to maximize hydrogen peroxide penetration without compromising its concentration in the gaseous phase. Special attention should be given to the effect of air compression in the middle of the lumen as a worst-case condition. On the other hand, the direct injection of a sterilant into medical devices emerges as a promising approach. However, it is of utmost importance to exercise special care and attention to detail when sterilizing the contact point of the injection port with medical instruments. This meticulous approach ensures the effectiveness and safety of the sterilization process, safeguarding the integrity of the medical devices and maintaining their functionality. In addition to research efforts, collaborations between VHP sterilizer designers and medical device and packaging designers can play a vital role in enhancing VHP penetration into medical devices and packaging materials. By jointly optimizing designs and considering the unique requirements of both sterilization processes and the devices themselves, these partnerships can pave the way for improved sterilization efficacy and safety. Through close collaboration, manufacturers and designers can address challenges related to VHP concentration, pressure alterations, and the injection of sterilant into medical devices, ultimately driving advancements in the field.

In conclusion, continuous research and development efforts in VHP sterilization cycle technologies hold great promise for enhancing the sterilization of reusable medical devices. By addressing challenges related to hydrogen peroxide penetration inside diffusion-restricted environments like lumens, researchers can drive significant improvements in sterilization outcomes and contribute to advancements in the field. Rigorous testing and validation studies, in line with regulatory guidelines, are crucial to ensure the reliability and effectiveness of proposed technologies in real-world applications.

## Figures and Tables

**Figure 1 microorganisms-11-02566-f001:**
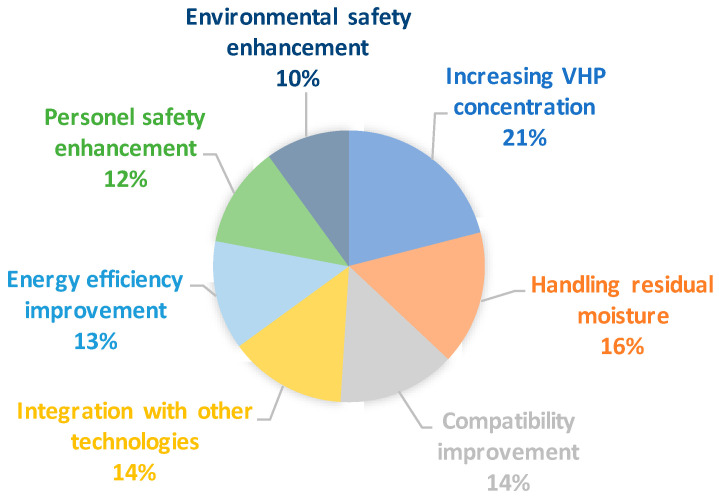
Key trends and innovations in VHP sterilization U.S. patent applications.

**Figure 2 microorganisms-11-02566-f002:**
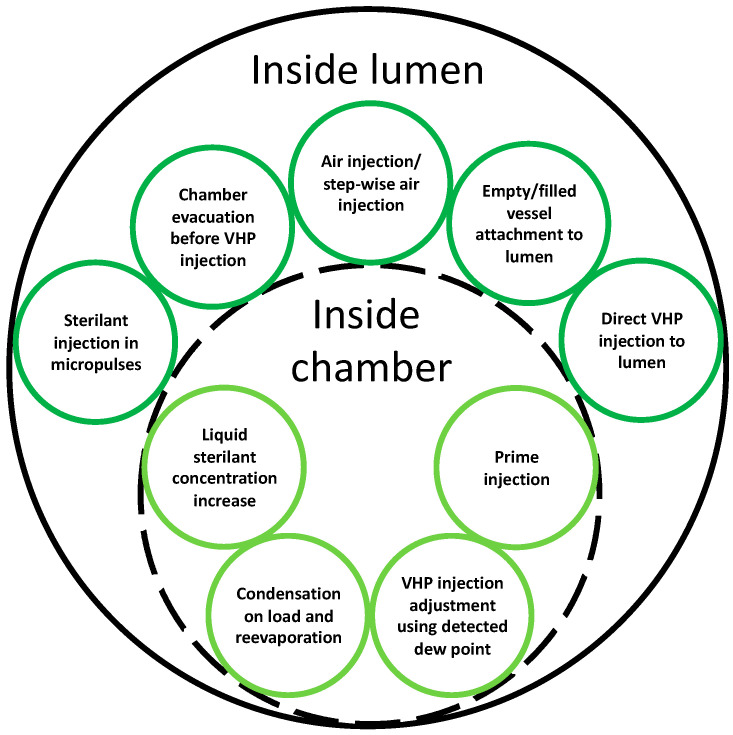
Different approaches to increasing the concentration of VHP inside a chamber or lumen.

**Figure 3 microorganisms-11-02566-f003:**
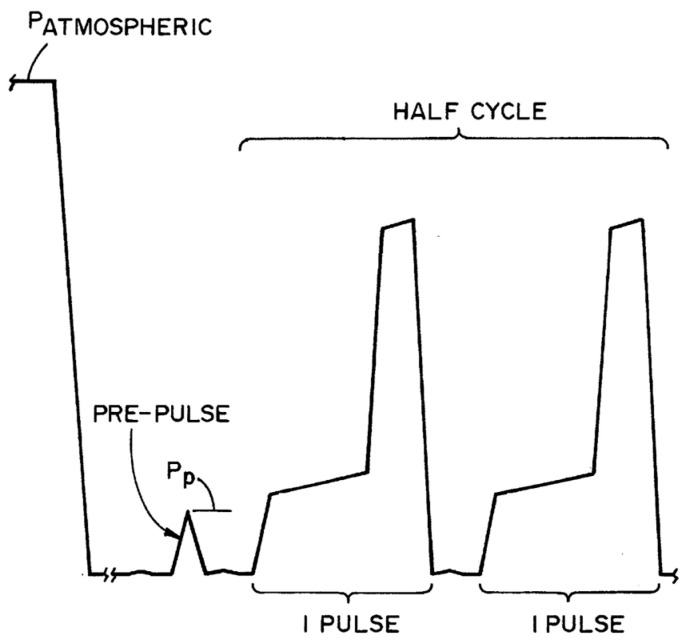
Demonstration of injection of sterilant into the chamber and then its evacuation to a vacuum pressure during the conditioning step (P_p_, pre-pulse, or prime injection) [26].

**Figure 4 microorganisms-11-02566-f004:**
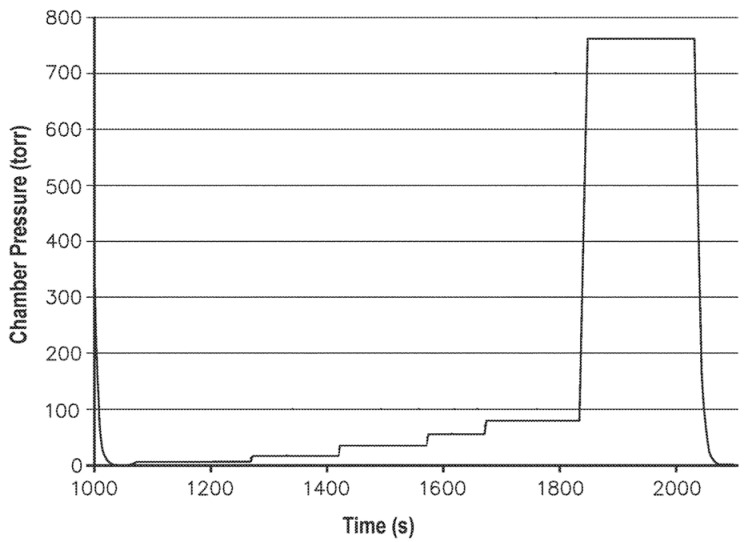
Incremental increase in the sterilization chamber pressure to provide a step-wise transition from vacuum to atmospheric pressure after VHP injection [28].

**Figure 5 microorganisms-11-02566-f005:**
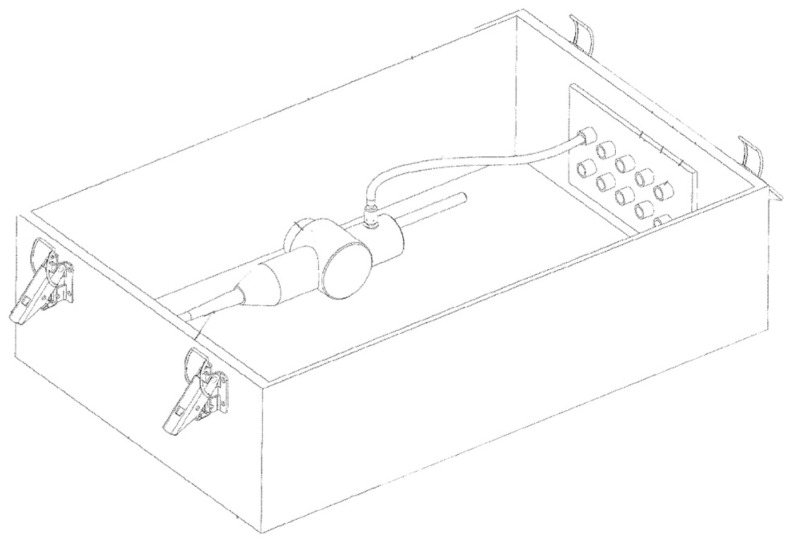
An example of direct injection of VHP into medical devices containing lumens using tubes connected to the sterilizer/packaging’s VHP injection port [32].

**Figure 6 microorganisms-11-02566-f006:**
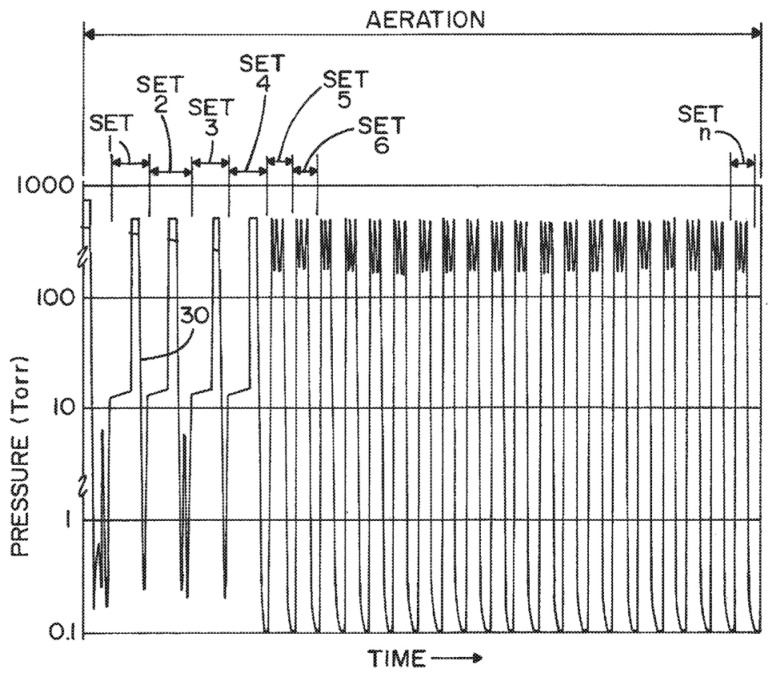
Using warm air pulses during the venting step to eliminate residual moisture at the end of the VHP sterilization cycle [40].

**Table 1 microorganisms-11-02566-t001:** Comparison of different approaches to detect the presence of initial residual moisture using chamber pressure analysis.

Detection Parameter	Advantages	Limitations	Reference
Pressure increase at a constant vacuum pressure along with gas concentration monitoring	-Distinguishing moisture from a virtual leak	-Non-efficient in detecting liquids other than water (like alcohols) -Reduced effectiveness if ice is formed -Requires water concentration sensor and high limit of detection of water vapor	[34]
Evacuation time difference/gas temperature difference	-Eliminating some parasitic parameters	-Non-sensitive to low amounts of water -Requires additional temperature sensor	[35]
A local maximum in pressure vs. time diagram during conditioning vacuum	-Distinguishing moisture on metallic vs. polymeric load -Eliminating some parasitic parameters	-Non-sensitive to low amounts of water -Requires high frequency and resolution of pressure data recording	[12]
A local maximum in the second-derivative of pressure vs. time diagram during conditioning vacuum	-Distinguishing moisture on metallic vs. polymeric load -Detecting low amounts of water	-Requires very high frequency and resolution of pressure data recording	[33]

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
