# Peer review of "Advances in Vaporized Hydrogen Peroxide Reusable Medical Device Sterilization Cycle Development: Technology Review and Patent Trends"

_microorganisms, 2023, doi:10.3390/microorganisms11102566_

Round 1
Reviewer 1 Report
Comments to the Author
In general, this article has met all criteria required by the journal. Explanation and discussion of the results are presented clearly enough and not confuse the reader. The results can give a positive value in advances in vaporized hydrogen peroxide reusable medical device sterilization cycle development. However, there are some issues that need further elaboration.
1. If more VHP is injected, will the economic efficiency improve? (Line120-Line122)
2. What is the basis for dividing the first and second half cycles in the process of material injection? (Line201-Line206)
3. Does congealed hydrogen peroxide have any sterilizing power? (Line220-Line224)
4. Apart from inflow and injection, there are no other methods for delivering sterilizers.
5. There are too few graphs to support the discussion adequately.
6. What does the error 'Reference source not found' mean? Should I consider deleting it? (Line350-Line351)
7. There are some grammatical errors, please check whole the manuscript for them.
OK
Author Response
1.1. If more VHP is injected, will the economic efficiency improve? (Line120-Line122)
Injecting more VHP in the sterilization chamber can have both advantages and disadvantages, and its impact on economic efficiency depends on various factors. For example, this improves the microbial efficacy, enhances sterility in diffusion-restricted environments, and reduces sterilization time. On the other hand, injecting more VHP negatively affects instrument compatibility, leads to residual sterilant concerns, increases the cost of sterilant consumption, and negatively impacts the environment.
Whether injecting more VHP improves economic efficiency depends on a careful balance between the benefits of improved sterilization efficacy and the potential drawbacks related to instrument compatibility, safety, and operating costs. In order to make this clearer for reader, the following sentence was added to the manuscript:
“A thorough cost-benefit analysis and consideration of specific application requirements are essential in determining the economic feasibility of increasing VHP concentration in a sterilization cycle.”
1.2. What is the basis for dividing the first and second half cycles in the process of material injection? (Line201-Line206)
Thanks for this question. The following paragraph has been added to the manuscript for more clarity:
In VHP sterilization devices, the sterilization cycle is typically composed of two identical half-cycles. The use of these half-cycles is often based on a method known as the "Overkill Approach", as described in the ISO14937:2009 standard. In this approach, a routine sterilant dwell period is extrapolated from the time required to achieve a 6-log reduction in a population of 10^6 biological indicators (BIs), ultimately providing a Sterility Assurance Level (SAL) of ≤10-6. This method assumes that the microbial inactivation kinetics follow a linear pattern, similar to what is observed in EOG sterilization of BIs like B. atrophaeus. It relies on the assumption that microbial populations treated with VHP will exhibit the same resistance to the applied lethal stress, as represented by a log-linear inactivation kinetic plot, often using populations like G. stearothermophilus or B. atrophaeus as BIs.
1.3. Does congealed hydrogen peroxide have any sterilizing power? (Line220-Line224)
Actually, the solid sterilant is not congealed hydrogen peroxide and this is composed of peroxide complexes like urea peroxide complex or sodium pyrophosphate peroxide complex. The sentence was modified and more explanation was added with reference for clarification:
The vessel could be either empty or filled with liquid or solid (urea peroxide complex or sodium pyrophosphate peroxide complex [23]) sterilant.
1.4. Apart from inflow and injection, there are no other methods for delivering sterilizers.
VHP sterilization devices, such as STERRAD, V-Pro, PSD-85, and STERIZONE VP4, primarily use a liquid sterilant, evaporate it using an evaporator, and then inject it into the chamber as the primary method for delivering sterilizers. These systems are indeed widely used for medical device sterilization and have been approved by the FDA for sale in the US market. There is no other commercial device approved by the FDA for sale in the US market and in the scope of this manuscript.
1.5. There are too few graphs to support the discussion adequately.
Thanks for this comment. To increase the clarity of discussions five new figure have been added to the revised manuscript.
1.6. What does the error 'Reference source not found' mean? Should I consider deleting it? (Line350-Line351)
Thanks for your comment. This has been a problem with word hyperlink for Table 1. The error has been solved and Table 1 was added to the text.
1.7. There are some grammatical errors, please check whole the manuscript for them.
The manuscript has been thoroughly inspected and some grammatical errors have been solved.
Reviewer 2 Report
The outcome of this work is interesting. However, the manuscript needs some corrections. A major revision is required before the manuscript can be considered for publication in Microorganisms. Some specific comments are listed to the authors for potential reference.
1. While the authors mentioned that there are no substantial safety concerns, could you elaborate on the safety measures and precautions in place to ensure the safety of patients, personnel, and the environment during VHP sterilization? Highlight and add discussion.
2. How effective is VHP sterilization in eliminating pathogens and ensuring the safety of medical instruments, especially when compared to other sterilization methods? Highlight and add discussion.
3. What specific advantages does VHP sterilization offer in terms of speed and energy efficiency compared to traditional sterilization methods for temperature-sensitive instruments? Highlight and add discussion.
4. The review has to be enhanced by adding enough recent references. The authors can follow the following references (https://doi.org/10.1016/j.ajic.2021.06.012); (https://doi.org/10.1007/s11356-022-21160-7); (https://doi.org/10.2345/0899-8205-54.s3.74).
5. What are the key trends and innovations identified in US patent applications related to VHP sterilization? Are there any noteworthy inventions or developments that stand out?
6. Could the authors provide more details on how VHP sterilization addresses the challenge of handling residual moisture, and why this is important?
7. Highlight and add more details about the existing IP-free technologies in VHP sterilization, and how do they contribute to the field?
8. Are there specific regulatory requirements or standards that VHP sterilization must adhere to in the context of medical devices, and how are these met? Highlight and add discussion.
9. Include additional descriptive figures or schemes to enhance reader engagement and comprehension.
Author Response
1.1. While the authors mentioned that there are no substantial safety concerns, could you elaborate on the safety measures and precautions in place to ensure the safety of patients, personnel, and the environment during VHP sterilization? Highlight and add discussion.
Vaporized Hydrogen Peroxide (VHP) sterilization ensures safety through its environmentally friendly, biodegradable nature, compatibility with most medical materials, and minimal risk to healthcare workers and patients. Unlike some other methods, VHP breaks down into harmless water vapor and oxygen, leaving no harmful residues or environmental impact. Its proven compatibility with medical instrument materials minimizes damage risk, while its quick sterilization process reduces personnel exposure and enables rapid instrument turnaround. The bellow sentence was added to the revised manuscript for suggestions for safety measures and precautions in place to ensure the safety of patients, personnel, and the environment during VHP sterilization:
"Stringent regulatory compliance, precise monitoring, and comprehensive training further bolster safety, making VHP sterilization a trusted and efficient method for safeguarding patients, personnel, and the environment in healthcare settings."
1.2. How effective is VHP sterilization in eliminating pathogens and ensuring the safety of medical instruments, especially when compared to other sterilization methods? Highlight and add discussion.
Thanks for your comment. The following paragraph was added to the revised manuscript to discuss the effectiveness of VHP sterilization in eliminating pathogens and ensuring the safety of medical instruments, especially when compared to other sterilization methods:
"In general, VHP sterilization is effective, safe, compatible, fast, and energy efficient, particularly when compared to other sterilization methods. VHP's efficacy is attributed to its ability to rapidly penetrate even diffusion-restricted environments, effectively killing microorganisms. Compared to traditional methods like steam sterilization, which cannot be used for temperature-sensitive instruments like endoscopes, VHP offers a versatile solution. It excels in achieving a high level of microbial reduction while also being compatible with most polymeric materials. Furthermore, VHP's biodegradable by-products and minimal environmental impact make it a safer choice. Compared to alternatives like EOG, which can leave hazardous residues and require extended ventilation periods, VHP's quick sterilization cycle and reduced exposure time enhance both instrument safety and workflow efficiency."
1.3. What specific advantages does VHP sterilization offer in terms of speed and energy efficiency compared to traditional sterilization methods for temperature-sensitive instruments? Highlight and add discussion.
The following discussion was added to the revised manuscript to discuss specific advantages that VHP sterilization offer in terms of speed and energy efficiency compared to traditional sterilization methods for temperature-sensitive instruments:
"Furthermore, VHP can complete the sterilization process in less than one hour, making it notably faster than methods like steam sterilization, which require longer cycles or EOG sterilization which required ventilation for several hours. Additionally, VHP operates at lower temperatures, reducing energy consumption and making it an energy-efficient choice in comparison to steam sterilization which requires evaporation of extensive amount of water and EOG which needs several hours of ventilation. On the other hand, due to significant constraints like scale, penetration, and compatibility with packaging materials, the widespread adoption of VHP sterilization for single-use devices has yet to gain traction. Nevertheless, recent advancements in sterilization chamber design and cycle development offer a fresh perspective and potential for exploration [7]."
1.4. The review has to be enhanced by adding enough recent references. The authors can follow the following references (https://doi.org/10.1016/j.ajic.2021.06.012); (https://doi.org/10.1007/s11356-022-21160-7); (https://doi.org/10.2345/0899-8205-54.s3.74).
The scope of this manuscript is to delve into recent advances in “VHP sterilization cycle development”. While I appreciate the provided references, it's important to clarify their relevance to our specific scope:
- The reference (https://doi.org/10.1016/j.ajic.2021.06.012) primarily compares the effectiveness of VHP sterilization with other methods, but it does not delve into the intricacies of cycle development, which is the central theme of our work.
- The second reference (https://doi.org/10.1007/s11356-022-21160-7) focuses on plasma disinfection, which falls outside the scope of this manuscript which is sterilization. On the other hand, although we briefly mentioned the combination of VHP and plasma in our work, the core focus remains on VHP sterilization cycle development.
- The third reference (https://doi.org/10.2345/0899-8205-54.s3.74) is indeed relevant and valuable to our research. I have incorporated discussions and insights from this review paper into the revised manuscript to enhance its comprehensiveness, as copied bellow.
I appreciate the suggestions and have taken steps to ensure that the manuscript remains aligned with its primary objective of exploring recent advancements in VHP sterilization cycle development through a patent analysis.
1.5. What are the key trends and innovations identified in US patent applications related to VHP sterilization? Are there any noteworthy inventions or developments that stand out?
The following paragraph has been added to the revised manuscript to explain about key trends and innovations identified in US patent applications related to VHP sterilization.
The examination of US patent applications on VHP sterilization revealed that most of the patents in this area aim at increasing VHP concentration, handling residual moisture, compatibility improvement, integration with other technologies, energy efficiency improvement, personnel safety enhancement, and environmental safety enhancement. Particularly notable inventions include precise control of VHP concentration and inventive moisture removal techniques, underscoring a collective commitment to optimizing sterilization processes efficiency and safety. In addition, the patens on this cate-gory generally claim modifications in the sterilization cycle, which is the main aim of this review. Therefore, this review is focused on the patents aim to increase VHP concentration and handle residual moisture.
1.6. Could the authors provide more details on how VHP sterilization addresses the challenge of handling residual moisture, and why this is important?
The following paragraph has been added to the revised manuscript to show the importance of challenges in handling residual moisture.
Handling residual moisture in VHP sterilization is paramount as it ensures the effectiveness and safety of the sterilization process. This is essential because moisture can impede the penetration of VHP, potentially leading to incomplete sterilization and posing risks to patient safety. Moreover, moisture can act as a protective shield for microorganisms, making effective sterilization challenging. Detecting moisture in a VHP sterilization cycle presents several challenges. Residual moisture, often hidden in diffusion-restricted areas of medical instruments, can interfere with the sterilization process, making accurate detection crucial. However, achieving precision in moisture measurement, especially in real-time, can be technologically demanding. Additionally, instrument design and material compatibility must be carefully considered to ensure reliable moisture detection. Innovative monitoring technologies continue to evolve to address these complexities in moisture detection.
1.7. Highlight and add more details about the existing IP-free technologies in VHP sterilization, and how do they contribute to the field?
The following paragraph has been added to the revised manuscript to give information about existing patents in VHP sterilization. The patens on increasing VHP concentration and handling residual moisture are well discussed in this work, while others are out of scope of this work.
The examination of US patent applications on VHP sterilization revealed that most of the patents in this area aim at increasing VHP concentration, handling residual moisture, compatibility improvement, integration with other technologies, energy efficiency improvement, personnel safety enhancement, and environmental safety enhancement. Particularly notable inventions include precise control of VHP concentration and inventive moisture removal techniques, underscoring a collective commitment to optimizing sterilization processes efficiency and safety. In addition, the patens on this cate-gory generally claim modifications in the sterilization cycle, which is the main aim of this review. Therefore, this review is focused on the patents aim to increase VHP concentration and handle residual moisture.
1.8. Are there specific regulatory requirements or standards that VHP sterilization must adhere to in the context of medical devices, and how are these met? Highlight and add discussion.
The following paragraph has been added to the revised manuscript to give information about regulatory requirements that VHP sterilization must adhere to in the context of medical devices:
VHP sterilization for medical devices is subject to rigorous regulatory requirements, particularly in the United States, overseen by the Food and Drug Administration (FDA). Compliance involves adhering to recognized standards such as ANSI/AAMI ST58:2013 and entails a meticulous validation process. Manufacturers must conduct validation studies demonstrating consistent microbial inactivation using biological indicators and routine monitoring of critical parameters. Continuous monitoring and validation are necessary to accommodate changes in cycle design, device design, and materials, ultimately safeguarding the safety and effectiveness of medical devices treated with VHP sterilization in healthcare settings.
1.9. Include additional descriptive figures or schemes to enhance reader engagement and comprehension.
Thanks for this comment. To increase the clarity of discussions, five new figure have been added to the revised manuscript.
Reviewer 3 Report
In this paper, the authors did a review of the vaporized hydrogen peroxide (VHP) terminal sterilization for the application of temperature-sensitive medical instruments. I recommend resubmission of this work after addressing the following issues. I will re-evaluate the suitability for publication.
1. The authors mentioned “temperature-sensitive instruments like endoscopes made of polymeric parts”. Firstly, please provide reference support. Secondly, are there any other examples of temperature-sensitive instruments? Thirdly, polymeric parts do not necessarily be temperature-sensitive at 121 °C. More attention is needed to make this statement more precise.
2. For “Analysis of patent trends”, it would be helpful if the author can include a (hierarchical) schematic to illustrate different categories involved.
3. Why would the authors like to put Table S2 in the supplementary?
4. The author mentioned “Some sophisticated approaches like incorporating air plasma” in Line 498. Actually, air plasma is commonly used for surface cleaning and disinfection. Please add this point as well with a reference support from a materials research paper: doi.org/10.1039/C9CC02967B.
5. The section “3. Analysis of 510(k) premarket notification” is still weak. I encourage the authors to add more discussions about the evidence provided.
6. Reference in “Line 350” reports error.
Author Response
1.1. The authors mentioned “temperature-sensitive instruments like endoscopes made of polymeric parts”. Firstly, please provide reference support. Secondly, are there any other examples of temperature-sensitive instruments? Thirdly, polymeric parts do not necessarily be temperature-sensitive at 121 °C. More attention is needed to make this statement more precise.
- Regarding the statement about "temperature-sensitive instruments like endoscopes made of polymeric parts," it is a general observation based on the well-established fact that endoscopes often contain delicate optical and polymeric components that can be adversely affected by high temperatures. This observation is widely recognized in the medical field due to the potential for heat-induced damage to endoscope components at temperatures higher than 60 °C. A reference has been added to text in the revised manuscript to support this statement.
- There are indeed other examples of temperature-sensitive medical instruments. These may include other types of surgical instruments with delicate components like electronic medical devices and equipment with sensitive sensors, all of which can be adversely affected by high-temperature sterilization methods. This example has also been added to the revised manuscript.
- Your point about the precision of the statement is valid and not all polymeric materials are temperature-sensitive at 121°C. The sensitivity of polymeric materials can vary widely depending on their composition and intended use. But the VHP sterilization devices are well-known as effective devices for sterilization of flexible endoscopes which often cannot tolerate temperature above 60 °C. This statement with the corresponding reference also has been added to the revised manuscript for more clarity.
1.2. For “Analysis of patent trends”, it would be helpful if the author can include a (hierarchical) schematic to illustrate different categories involved.
Figure 1 has been added to the revised manuscript to show the key trends and innovations in U.S. VHP sterilization patent applications.
1.3. Why would the authors like to put Table S2 in the supplementary?
The table provides a summary of 510(k) premarket notification clearances for STERRAD 100NX and NX, as well as V-Pro Max and S series sterilizers, which are the only commercially available VHP sterilization devices in U.S. market. The primary purpose of this table is to analyze modifications made to cycles of these sterilizers. Importantly, the table demonstrates that there have been no significant changes in the cycle development, which is the specific focus of this work. As the table's content is not central to the main narrative and does not provide extensive information within the context of the manuscript's scope, it has been relocated to the supplementary information to maintain the manuscript's focus on the key trends and innovations in VHP sterilization “cycle” development.
1.4. The author mentioned “Some sophisticated approaches like incorporating air plasma” in Line 498. Actually, air plasma is commonly used for surface cleaning and disinfection. Please add this point as well with a reference support from a materials research paper: doi.org/10.1039/C9CC02967B.
Thank you for your thoughtful feedback and clarifications regarding the mention of "air plasma" in our manuscript. We appreciate your diligence in reviewing our work.
Upon careful consideration, we acknowledge that the context in which "air plasma" is mentioned is specifically related to the elimination of residual moisture during the VHP sterilization process, as indicated in the "Residual moisture elimination" section of the paper. It is indeed used as a technology to aid in the evaporation of residual moisture rather than for surface cleaning, disinfection, or sterilization.
Given the clarification you provided, it would not be in the scope of this work to refer the paper doi.org/10.1039/C9CC02967B (A self-assembled peptidic nanomillipede to fabricate a tuneable hybrid hydrogel), which is not in the scope of air plasma or disinfection.
1.5. The section “3. Analysis of 510(k) premarket notification” is still weak. I encourage the authors to add more discussions about the evidence provided.
As mentioned in the response to question 3, the aim of this section is to analyze 510(k) premarket notification clearances for STERRAD 100NX and NX, as well as V-Pro Max and S series sterilizers, which are the only commercially available VHP sterilization devices in U.S. The primary purpose of this table is to analyze modifications made to cycles of these sterilizers. Table S2 which is discussed int this section demonstrates that there have been no significant changes in the cycle development, which is the specific focus of this work. For sure other modifications (not related to cycle development) would be discussed in this section, but this is out of the scope of manuscript which is on the key trends and innovations in VHP sterilization “cycle” development. I believe showing there is no modification in the cycle is very interesting for authors and complementary of patent trends review. This section shows, as explained in the manuscript: “no changes have been observed on the cycles after their clearance which could be attributed to the enormous efforts required for verification and validation activities for numerous medical instruments”.
Some general text has been added to the section “3. Analysis of 510(k) premarket notification” of the revised manuscript, to give more information about 510(k) process concerns and importance of the data obtained from 510(k) analysis.
1.6. Reference in “Line 350” reports error.
Thanks for your comment. This has been a problem with word hyperlink for Table 1. The error has been solved and Table 1 was added to the text.
Round 2
Reviewer 1 Report
ok
ok
Author Response
Thanks for your time to evaluate my manuscript. The revised manuscript has been proofread once more and some parts has been improved.
Reviewer 2 Report
The overall organization of the manuscript continues to require substantial improvement. The flow of the content remains incoherent, which makes it challenging for readers to effectively follow the author's arguments. Additionally, some sections that were previously requested to be improved still appear incomplete, and key references relevant to the subject matter are still missing. Furthermore, the methodology and data analysis sections continue to lack the necessary detail and rigor expected for a review article. It is imperative to provide a clear and transparent description of the research methods employed and the criteria for selecting the studies included in the review.
Author Response
First and foremost, I would like to extend my appreciation for the time and effort you've invested in reviewing my manuscript. Your feedback is valuable, and I'm dedicated to ensuring the manuscript meets the high standards expected by the journal and its readers.
Regarding the concerns you've raised:
1- Organization and Coherence: I understand the concerns about the flow and organization of the content. However, please note that the manuscript is structured as a patent review, which inherently differs from traditional research review papers. As a result, the sequence and nature of content presentation are aligned with the guidelines and expectations specific to patent reviews.
2- Sections Needing Improvement & Missing References: I have taken efforts to incorporate feedback from the initial round of reviews and re-examined the sections you've highlighted. Additionally, I reviewed and ensured the inclusion of all pertinent references to strengthen the manuscript.
3- Methodology and Data Analysis: The format of a patent review often varies from standard review articles, especially regarding methodologies and data analysis. Nevertheless, I attempted to provide a clearer and more comprehensive description, ensuring it remains consistent with the patent review structure and doesn't detract from the primary focus.
In light of your feedback, I revisited the manuscript to make the necessary revisions, ensuring it remains true to its nature as a patent review while addressing the concerns you've highlighted. I truly value your insights and worked diligently to improve the quality and clarity of the content.
Thank you again for your thorough feedback, and I hope the revised manuscript will meet your expectations and those of the journal.
Reviewer 3 Report
I suggest acceptance of this work in current form since the authors have addressed my comments.
Author Response
Thanks for your time and evaluating the revised manuscript.